# Impact of COVID-19 on School Populations and Associated Factors: A Systematic Review

**DOI:** 10.3390/ijerph19074024

**Published:** 2022-03-29

**Authors:** Andi Muhammad Tri Sakti, Siti Zaiton Mohd Ajis, Arina Anis Azlan, Hyung Joon Kim, Elizabeth Wong, Emma Mohamad

**Affiliations:** 1Centre for Research in Media and Communication, Faculty of Social Sciences and Humanities, Universiti Kebangsaan Malaysia, Bangi 43600, Malaysia; andi.muhammad@mercubuana.ac.id (A.M.T.S.); emmamohamad@ukm.edu.my (E.M.); 2UKM x UNICEF Communication for Development Centre in Health, Faculty of Social Sciences and Humanities, Universiti Kebangsaan Malaysia, Bangi 43600, Malaysia; zaiton2007@gmail.com; 3Faculty of Communication Science, Mercu Buana University, Jakarta 11650, Indonesia; 4UNICEF Malaysia Country Office, Putrajaya 62100, Malaysia; hkim@unicef.org (H.J.K.); elwong@unicef.org (E.W.)

**Keywords:** health crises, COVID-19, school populations, systematic review

## Abstract

Apart from the severe impact on public health and well-being, the chain effect resulting from the COVID-19 health crisis is a profound disruption for various other sectors, notably in education. COVID-19 has driven massive transformation in many aspects of the educational landscape, particularly as teaching and learning shifted online due to school closure. Despite the many impacts of the health crises on school populations, a systematic review regarding this particular issue has yet to be conducted. This study, therefore, attempts to comprehensively review the impact of health crises on school populations (student, teacher, parent, and school administration). An extensive literature search guided by the Preferred Reporting Items for Systematic Reviews and Meta-Analysis (PRISMA) reporting checklist was performed in two selected databases, namely Web of Science (WoS) and Scopus to identify how this particular topic was previously studied. Exclusion and inclusion criteria were set to ensure that only research papers written in English from the year 2000 to the present (April 2021) were included. From a total of 457 studies screened, only 41 of them were deemed eligible to be included for qualitative synthesis. The findings revealed that the COVID-19 pandemic was the only health crisis discussed when it comes to investigating the impact of health crises on school populations. This study found four notable consequences of health crises on school populations, which are impacts on mental health, teaching and learning, quality of life, and physical health. Among factors associated with the impact of the health crises are; demographic factors, concerns about the pandemic, education-related factors, health-related factors, geographic factors, economic concerns, teaching challenges, and parenting in the pandemic. This study is expected to be a reference for future works in formulating crises mitigation strategies to reduce the impact of health crises on schools by exploring the contexts of the crises.

## 1. Introduction

Health crises, also termed as a health emergency, can be constituted by emergent events such as devastating infectious disease and natural disasters (e.g., earthquake, flood, and wildfire) [1] whose scale, timing, or unpredictability threatens to overwhelm routine capabilities after resulting in negative health consequences to the community [2]. Previous work corroborates the above definition, highlighting that response capacities are overburdened due to urgency of response and high uncertainty in a health crisis [3]. In cases of immediate health threats such as viral pandemics, hurricanes, and tornadoes, authorities will commonly issue an emergency declaration to mitigate the impact of the crises on society, in which funds, personnel, and all needed resources will be deployed as an aid for strategic intervention [4].

Apart from its severe impact on public health and well-being, the chain effect resulting from health crises has profound effects on various other sectors, notably on education. As an example, the destruction of public places and houses followed by the reduction of income in the aftermath of natural disasters has brought difficulty for low-income households. This group is perceived to have low resources for loss recovery and may not prioritize health and education in rebuilding after a disaster [5]. Another example can also be drawn from viral epidemic cases like influenza. Considering young school-age children as the most vulnerable group to this particular disease [6], the implementation of school closure has been outlined as the best mitigation strategy for this health crisis in many previous studies. De Luca et al., through their mathematical approach to student contact patterns, revealed that school holidays contribute to delaying the peak of influenza in the Belgian school population [7]. In this case, school closure is considered to be the most effective intervention strategy because of its contribution to reducing contact and virus transmissibility, especially when attack rates were found to be higher in children compared to adults [8].

Presently, the world is threatened by the deadly COVID-19 virus which has resulted in over five million deaths in the global population [9] within the two years since its first emergence, in late December 2019. The unprecedented change caused by this pandemic has also driven a massive transformation in many aspects of the educational landscape, particularly as teaching and learning was moved online due to school closure. This is especially true in fields where in-person interaction is requisite, for instance, field education, vocational education and training, and medical education. A study conducted by Dempsey et al. (2021) outlined the difficulty faced by the faculty during the earlier phase of the COVID-19 pandemic in managing field education curriculum and maintaining mutual relationships with the agencies while having to deal with the fear of the unknown [10]. In the vocational education and training field, the disruption mostly occurred when schools were forced to provide hands-on practical training using virtual tools and machines through online communication platforms [11]. Considered as the most affected area among all educational fields, medical education had to pause its most essential practical learning domain, i.e., clinical learning experience [12]. In Malaysia, teacher preparedness and competency in conducting virtual learning and student access to virtual learning were still major issues that have yet to be solved [13]. A prior study involving school teachers in Malaysia also concluded several challenges faced by them in conducting online learning such as work and information overload, unfamiliarity with new online learning platforms, and personal health challenges related to stress and anxiety [14].

More worryingly, it has been proven the effect of the pandemic negatively affects the whole school ecosystem including its main populations—from students, teachers, administration, to parents. One of the most highlighted impacts of the COVID-19 pandemic on these populations is its effect on the mental health of students, parents, and teachers. A recent study involving university students in the United States found that most of the respondents experienced an increase in stress and anxiety due to the pandemic and school closure, and difficulty in concentrating was found to be the second-highest factor contributing to that stress [15]. The impact of COVID-19 on parents’ mental health was also discussed, in which one study suggested that parents with school-age children demonstrated poorer mental health conditions compared to parents without school-age children at home [16]. School teachers also experienced stress related to the change to remote teaching and emotional exhaustion when teaching during the pandemic [17].

Despite the negative effects of health crises on the school ecosystem, especially among its populations, a systematic review regarding this issue has yet to be conducted. Prior to COVID-19, systematic review studies have outlined the impact of viral epidemics such as SARS, H1N1 influenza, and Ebola [18] and endemics like Malaria [19]. However, neither studies examined the impact on schools, but rather addressed the impact on public health workers. In March of 2021, Hosseinighousheh et al. summarized the impact of natural disasters on schools, but the scope of the review was limited to studies that involved the Iranian school population [20]. Since COVID-19 has become the epicenter of public health discussion in recent years, scholarly attention has been drawn to review the impact of this infectious disease on schools. Most of these reviews summarized the impact of COVID-19 on students, in particular they have identified that student health was the most prominent issue including mental health [21,22,23,24], physical health [25], or both mental and physical health [26].

This study, therefore, will seek to provide a comprehensive analysis of the impact of health crises on all major populations in the education system such as students, teachers, parents, and school administration. This paper will review the type of health crisis, the impact of health crises, the population studied, as well as the approach used to study the issue. The objectives of this study are; (i) to map the impact of health crises among school populations all over the world, and (ii) to identify factors contributing to the impact of the health crises on school populations. The results of this study are expected to be a reference for future work in formulating crises mitigation strategies to reduce the impact caused by health crises on schools by learning from the contexts within which the crises occur.

## 2. Materials and Methods

A systematic review was conducted to summarize the impact of health crises on school populations as well as the contributing factors surrounding those impacts. In identifying how this particular topic was studied in prior literature, an extensive literature search guided by the Preferred Reporting Items for Systematic Reviews and Meta-Analysis (PRISMA) reporting checklist [27] was performed in two selected databases, namely Web of Science (WoS) and Scopus. The protocol for this review was not registered to a systematic review database.

### 2.1. Eligibility Criteria

A set of pre-defined criteria for inclusion eligibility were decided to assess the suitability of articles to be included in the review process. The first criterion was the year of publication, which was set to include all articles from January 2000 to the present (April 2021). The second criterion was that the article had to be an empirical research paper written in English, any other document type such as reviews, editorials, notes, or book chapters were then excluded. In terms of source, it was decided that only articles published in journals would be included. Therefore, papers from conference proceedings, books, or book series were deemed ineligible to be reviewed. This decision was made to ensure the work selected was of high quality (i.e thorough analysis, peer-reviewed) with adequate information for reviewers to evaluate [28]. Papers in the social sciences, arts and humanities, and hard sciences were included. The last and the most important criterion was that the article had to discuss the impact of health crises on school populations. To pass the eligibility for inclusion on the third criterion, the article did not only need to mention specific impacts of health crises but most importantly, had to address the impact of health crises on school populations as the main discussion of the research. Table 1 below presents the inclusion and exclusion criteria of the search through databases.

### 2.2. Systematic Review Process

There were several stages included in the systematic review process, starting with a literature search on the Scopus and Web of Science (WoS) databases on 16 April 2021. The search string was developed by including keywords related to health crises, impact, and school populations, so that any article with these specific keywords in its title, abstract, or abstract keywords, would be retrieved by the system. All retrieved articles proceeded to the next stage: title and abstract screening. To aid in the title and abstract screening, the articles and abstracts were exported to a .csv file so that screening could be performed in Microsoft Excel. The details of the search string is shown in Table 2 below.

A total of 282 and 385 articles were retrieved from WoS and Scopus databases respectively. The title and abstract screening detected redundancies between these two databases and as a result, 210 duplicates were removed, resulting in only 457 articles proceeding to the next stage in the screening process. A peer-review approach involving all authors [29] was used in the quality assessment process. In this phase, the researchers read the title and abstract of each article to decide which of them could proceed to the full-text screening stage. In the full-text screening stage, all selected articles were read thoroughly and only articles that received reviewer consensus on eligibility would be included in the qualitative synthesis. From 457 articles reviewed in the title and abstract screening, only 66 reached reviewer consensus for extensive full-text reading. A total of 391 articles were excluded as the titles and abstracts did not contain information that indicated a focus on impacts of health crises and their associated factors. Finally, the whole screening process resulted in only 41 articles deemed eligible for qualitative synthesis. Among the reasons for the exclusion of 25 articles from the qualitative synthesis were; the research was not conducted empirically (e.g., review or report) or the main focus of the research was not on the impact of health crises on school populations. For example, a study by Veenema et al. was excluded because the study only discussed nursing school and nursing educator preparedness for radiation emergencies [30], while the impact of nuclear events on the school population was not discussed or measured in the study. The flow of this systematic review process is explained in Figure 1.

### 2.3. Data Abstraction and Analysis

All 41 articles that proceeded to the full-text screening stage were thematized qualitatively through a content analysis method. A semi-structured coding process was conducted by utilizing Atlas.ti version 8 to categorize information contained in the papers into specific themes. The themes were constructed according to the objectives of the systematic review, which were to identify the impact of health crises and factors contributing to the impact of health crises in school populations. Additional themes such as type of health crisis, countries where these studies were conducted, research methods utilized, and population of the studies were also observed to identify patterns in the studies. Sub-themes associated with each theme were also defined based on an inductive analysis of the full-text articles.

## 3. Results

This section summarizes the vetting process in this systematic review, discusses the main themes and characteristics of the selected studies, and presents findings according to the objectives of this review, which are to identify health crisis impacts on school populations and factors surrounding the impact of the health crises.

### 3.1. Characteristics of the Included Studies

Of the selected studies, eight were from China [31,32,33,34,35,36,37,38], four from the United States [39,40,41,42], three from Italy [43,44,45] and Spain [46,47,48] respectively, two from both United Arab Emirates [49,50] and the Philippines [51,52], while the rest were scattered across different countries [53,54,55,56,57,58,59,60,61,62,63,64,65,66]. A few of the papers were comparative studies that compared the impact of COVID-19 in multiple countries, such as China and Hong Kong [67], Italy and Portugal [68], Spain and Mexico [69], Spain and Peru [70], and Italy, Spain, and Ecuador [71]. Most of the studies were conducted quantitatively (*n* = 37 studies), three studies used qualitative methods, and one study used mixed methods, with populations varying from students (*n* = 34 studies), teachers/lecturers (*n* = 5 studies), parents (*n* = 2 articles), and one study that compared the impact of COVID-19 between student and school administration populations. Table 3 summarizes these results.

### 3.2. Health Crisis Impact on School Populations

The qualitative analysis on the 41 selected studies divided the impact of health crises on school populations into four general categories. There were 27 articles that discussed the impact of the COVID-19 pandemic on mental health, 14 articles on teaching and learning, two articles on quality of life, one article on physical health, and one article on mental health and teaching and learning combined. There were two studies summarizing the impact of health crises on mental health and teaching and learning, and one study on mental health and physical health combined.

Interestingly, all eight studies originating from China and a comparative study involving China and Hong Kong populations consistently outlined the impact of COVID-19 on the mental health of students. Chen et al. in their cross-sectional study confirmed that the prevalence of depression and anxiety among Chinese high school students significantly increased after the initial outbreak, and students’ concerns related to entering a higher grade was positively associated with depression and anxiety [31]. Similarly, depression and anxiety were also reported among college students in China, where the stress of online learning and entering a higher grade were among the main contributing factors for the occurrence of depressive symptoms. Among Chinese university students, disruption of academic activities, severe economic impact, and health conditions were identified as predictors of poor mental health status. According to the findings of these studies, the most prevalent factor that affected the mental health of students was their fear of the potential impact of COVID-19 on their academic-related activities. Fear of the COVID-19 impact on academic activities was also reported among students in other countries such as the United States [39], United Arab Emirates [50], and Sudan [56].

COVID-19 preventive measures such as home quarantine also resulted in impacts on physical health. Through a qualitative study, de Maio Nascimento found that aside from increasing anxiety due to tragic news related to COVID-19, the group under study also reported experiencing physical-related issues such as sleeping disorders, loss of appetite, joint pain, weight loss, and reduction of functional capacity [64].

Among the selected studies, the second most discussed impact was on teaching and learning (*n* = 14 studies). Our review suggests that teaching and learning issues such as learning disruption, difficulty adapting to new teaching methodologies and change in learning platform were experienced by most university students (10 out of 14 studies), particularly in the United States (*n* = 3 studies). In a study conducted by Rana et al., US medical university students revealed that most of them were experiencing learning disruption from virtual class and modified schedules [41]. In another study by Hung et al., (2021) a majority of medical university students agreed that clinical education has suffered after transitioning online, as it disrupted on-site clinical activity [40]. Therefore, most university students, particularly those in medical programs, believed that the implementation of control measures such as social distancing, online learning, and clinical rotation during the pandemic would potentially obstruct the future development of medical education [40].

Aside from disrupting learning activities, the COVID-19 pandemic has also decreased student academic motivation. Through a study examining the impact of COVID-19 lockdowns on primary school students in Italy and Portugal, Zaccoletti et al. noted a pronounced decrease in academic motivation among the two studied populations [68]. The factors contributing most to poor academic motivation were low participation in extracurricular activities and age (younger students were less motivated). These findings reflect the findings of another study carried out by Fontanesi et al. which outlined parental burnout among Italian parents whose children were being homeschooled [44]. The study reported that the majority of parents noted a significant change in their child’s behavior such as their inability to concentrate, intolerance, and general discomfort. These two studies confirmed the vulnerability of both children and parents in managing academic-related activities while undertaking pandemic control measures.

In an attempt to cope with the negative outcomes of the COVID-19 pandemic, schools were also forced to be innovative to keep teaching and learning activities on track. For instance, in Sweden, Beery has proven the critical role of participatory risk management to mitigate the impact of COVID-19 on environmental and outdoor education [53]. This approach was found to be effective in managing the expected risk of COVID-19 infections while conducting activities in outdoor adventure settings. Another example can also be drawn from one school in Spain, where an Intelligent Personal Assistant (IPA) was developed and stored on a learning management system to facilitate self-regulated learning among students during school closure [48]. As a result, despite the need for improvement in information availability and feature usability, the IPA innovation succeeded in creating greater functionality in access to the learning management system and increased student satisfaction towards the education delivery in that school. The use of virtual learning methods indeed has been considered a great opportunity for teaching and learning development. Therefore, the change in education delivery, wherein virtual learning will take precedence over face-to-face learning, appears unavoidable post-pandemic since many schools have considered continuing with this approach when the crisis ends [55].

International higher education was predicted to be heavily impacted post-pandemic. Mok et al. identified that the current health crisis circumstances have remarkably decreased Chinese and Hong Kongese students’ willingness to pursue higher education abroad [67]. This situation has not only created a significant downturn in international student mobility but has also shifted the mobility flow itself, causing changes in popular study destinations. In consequence, many higher education institutions, particularly those that are dependent on international student tuition will suffer a great financial loss henceforward.

Lastly, the COVID-19 pandemic phenomenon has also brought negative implications for teachers. Apart from impacts on the mental health of teachers, teaching in a pandemic has been proven to contribute to teaching disruption and a decrease in quality of life. While assessing Filipino teachers’ confidence and competence in conducting virtual learning, Race identified that many teachers were unable to regularly participate in online classes due to a lack of facilities and resources [52]. Additionally, nearly all respondents believed that online class reduced teaching and learning participation rates, limited interaction and socialization, and was overall ineffective compared to face-to-face learning. Another study addressing the mental health impacts of COVID-19 among teachers in the Philippines reported moderate levels of quality of life within six months after national lockdown [51]. In Chile, most teachers reported experiencing a significant decrease in their quality of life. These three studies simultaneously suggested the need to address the psychological well-being of teachers to help them teach effectively in pandemic settings.

### 3.3. Factors Associated with the Impact of Health Crises

Factors associated with the impact of health crises on school populations were classified into nine main categories. The most prevalent factors associated with health crises in school populations were demographic factors (*n* = 13 studies), followed by concerns about the pandemic (*n* = 12 studies), education-related factors (*n* = 12 studies), health-related factors (*n* = 7 studies), geographic factors (*n* = 3 studies), economic concerns (*n* = 3 studies), teaching challenges (*n* = 1 study), and parenting in the pandemic (*n* = 1 study).

The most prominent factors were demographic factors. This review suggests that demographic factors such as gender, school grade, program, age, and residence have an impact on mental health (*n* = 10 studies), quality of life (*n* = 2 studies), and teaching and learning (*n* = 1 study) among school populations. In a comparative study assessing the impact of the COVID-19 pandemic on students and school administration, Marelli et al. found that anxiety is more prevalent and higher among students compared to members of the school administration [45]. In terms of gender, female populations (which in most studies were students) were reported to be more susceptible to psychological issues like depression and anxiety [31,45,49,58,63], stress [57,61], and suicidal ideation [59], while there was only one study which reported males as being more affected [32]. Place of residence was also among the demographic factors that predicted student mental health issues. Both studies conducted by Han et al. and Tasnim et al. suggested that living in urban areas was associated with higher anxiety and suicidal ideation respectively [32,59]. An interesting finding was presented by Drissi et al. where medical students were reported to perceive higher stress when engaging with activities in physical settings like hospital visits or clinical rotations which decreased with the introduction of online learning [49]. On the contrary, non-medical students experienced an increase in anxiety with online learning. Level of education and gender played a role in determining teacher quality of life, in which teachers enrolled in master’s programs were more affected by the pandemic compared to those who were enrolled in doctorate programs [51]. Female teachers experienced a greater impact on their quality of life compared to male teachers [60]. As for teaching and learning impacts, one of the reviewed studies revealed that younger students were prone to a decrease in academic motivation compared to older students [68].

All factors under the ‘concern about the pandemic’ theme found in this review were associated with mental health issues among university and college students. A majority of the student population in the study by Han et al. were reported to have a high level of the perceived risk of COVID-19 infection which made them reluctant and anxious to leave home [32]. The lack of government response and intervention worsened the situation and led the society to experience greater panic in the earlier phase of the pandemic [62]. Student concerns regarding COVID-19 preventive measures were also noted as significant predictors of mental health issues by some of the reviewed studies. Orders to remain at home with movement being restricted had increased negative emotions such as anxiety, depression, and ambiguity among college students [37,65], loneliness, depression, anxiety, stress, panic, despair, and suicidal ideation among university students [39,54,59] and also depressive symptoms among members of the school administration [45]. Meanwhile, fear of COVID-19 infection had increased student stress and depressive symptoms above the critical threshold [61], causing them to lose perceived control and experience pandemic-related difficulties [62], and negative emotions related to the COVID-19 death toll [38]. In other cases, this fear also decreased teacher quality of life [51,60].

The notable change in teaching and learning methods from face-to-face delivery to distance learning was also associated with the impacts of health crises discussed in this review. Distance learning was associated with psychosomatic problems, workload, and burnout among teachers [69], and sleep problems, stress, and other depressive symptoms among students [37]. Distance learning was found to significantly contribute to parental burnout and difficulties among parents in facilitating their children as most children indicated an inability to concentrate [44]. It was also observed that the change in learning platform was a factor that decreased student engagement and motivation [70], worsened their study [43], increased student workload [71], and limited teacher communication and socialization with their students [52]. However, distance learning has also brought an opportunity for schools to develop alternative solutions and innovations to optimize teaching and learning online, for instance, the invention of the Intelligent Personal Assistant (IPA) that was proven to increase student satisfaction toward virtual learning [48].

Another education-related factor found in this study is academic uncertainty (*n* = 3 studies), of which were all associated with the prevalence of mental health issues among students. According to the study conducted by Chen et al., the occurrence of depressive and anxiety symptoms among high school students was positively influenced by their concern about entering a higher grade [31]. Interestingly, the students worried about their academic uncertainties more than they worried about the high number of COVID-19 cases in their areas. Dangal & Bajracharya revealed that delays in academic delivery and the economic effects of the pandemic were associated with student anxiety symptoms [63]. The findings of this study were similar to Song et al. which suggested that academic uncertainty (i.e., difficulties to find a job and to study abroad) and economic pressure were associated with a greater Post-traumatic Stress Disorder (PTSD) score [34]. Both studies by Song et al. and Dangal & Bajracharya also identified economic concerns as supporting factors of the occurrence of mental health issues among students. In line with these findings, Cam et al. in their study assessing the impact of COVID-19 on Turkish university student mental health and quality of life, identified low family income, female gender, and poor family relationship as risk factors for depressive symptoms, stress, and anxiety among students during the pandemic [58].

The next prominent factor for health crises impact among school populations is health-related factors (*n* = 7 studies). Several reviewed studies outlined the contribution of physical and mental components to the occurrence of mental health issues during the COVID-19 pandemic. A study examining the traumatic effects of COVID-19 in China revealed that poor self-rated health status was highly associated with greater psychological impact and worse mental health among Chinese students in the earlier phase of the pandemic [34]. In Bangladesh, through their study measuring suicidal ideation among university students, Tasnim et al. divided the potential factors of suicidal ideation into several physical and mental health components such as less sleep, excess sleep, cigarette smoking, depression, stress, and anxiety [59]. Yu et al., in another study, confirmed that sleep problems were among the risk factors of depressive symptom prevalence among college students in China [36] while Yao Zhang et al. suggested that the lack of physical activity contributed to negative emotions [38].

All the geographic factors identified in this review were associated with mental health issues. In the earlier phase of COVID-19 emergence in Wuhan, Hubei Province, students who lived in that province were reported to experience an “emotional infection” and were more nervous and scared compared to those who lived outside the province due to an information exposure effect [37]. Mobility issues caused by the COVID-19 pandemic have also resulted in mental health impacts. Conrad et al. outlined that the relocation mandate that forced students to vacate on-campus residence and leave their personal belongings caused more experiences of COVID-19-related grief, loneliness, worries, anxiety, depression, and PTSD symptoms compared to those who were not mandated to leave the campus [39]. Song et al. also found that the COVID-19 disruption on mobility has caused students who were planning to study abroad to experience PTSD [34].

Although not as prominent compared to the factors discussed above, the occurrence of several other factors such as personality traits, teaching challenges and parenting in the pandemic were also identified as factors associated with the impact of health crises, particularly in mental health. Two studies have shown that students with certain personality characteristics such as introversion [32] and neuroticism [62] faced a greater risk of anxiety during the COVID-19 pandemic. Challenges faced by school teachers such as workload and the inequity of task distribution have become an important risk factor to the psychosocial well-being of teachers. Lastly, parenting during the pandemic has been proven to be a significant predictor of parental burnout.

A broad observation on trends shows that education-related factors were the factor most associated with the impact of COVID-19 in the year 2020, indicating that uncertainties related to academic activity, method transition, and future academic plans were of high concern at the time. That year, many education and health authorities enforced online learning as the main strategy to help schools avoid academic performance lag, which in many cases has led many students to experience a decrease in academic motivation and performance. Evidently, with schools reopening near the end of the year, research on this has become less prevalent in 2021.

## 4. Discussion

This paper is among the first systematic reviews on the impact of health crises on a broader school population including students, teachers, parents, and school administration. Other review studies on the impact of health crises, particularly on the COVID-19 pandemic, have focused on specific issues like mental health [72,73,74,75]. Cachón-Zagalaz et al. initiated a review of the impact of COVID-19 on school populations, but the studied population was limited to school children [76]. Thus, the results of this study are expected to offer more comprehensive understanding on multi-tier impacts on the school ecosystem caused by health crises.

Most studies in this review highlighted the issue of mental health which implies that the most significant impact of the COVID-19 pandemic on school populations is on mental health. This is evidenced by the prominence of mental health issues among students due to distance learning and a sudden transition of learning method [44,69,70]. Subsequently, this points to the need for mental health support in educational settings. Schwartz et al. suggested that physical school attendance is important for mental health and allows students to access mental health services [61]. Two options that may be considered in improving mental health during a health crisis are to (i) strengthen access to and quality of mental health services (online and offline); and (ii) prioritize making physical schooling safer to avoid disruption in education. A prior study carried out in Malaysia also outlined the importance of acknowledging the mental health issues among teachers and suggested the development of tele-counseling and other mental health advisory services to mitigate the mental health impact on the affected educators [77].

To strengthen access to and quality of mental health services in the education system during a health crisis, system-level change has to take place. Schools often struggle in allocating resources for support services such as counseling [78]. However, an emphasis on the mental well-being of the school community increases resilience to crisis situations [79]. Thus, the root of addressing mental health issues must involve systemic change. This involves policy and institutional support from the government or relevant authorities, private sectors, civil society and NGOs to provide access to mental health experts, improve referral pathways for seeking mental health support and create an inclusive environment for mental health in the school ecosystem. The role of mass and social media to consistently disseminate information on government aid and policy to mitigate the impact of health crises, particularly in mental health, should be strengthened to advocate this purpose [80], including reducing stigma and discrimination surrounding mental health.

While proper planning and implementation is a prerequisite, the literature highlighted that minimizing the disruption of physical learning during health crises helps reduce anxiety among students and ensure continuous learning. The challenge includes creating safe learning spaces which allow social interactions yet are able to minimize infection. This is expected to be among the most practical solutions to help teachers [52,71] and parents [68] deal with the disruptions and challenges they faced amid the implementation of distance learning.

Most of the reviewed studies placed the student as the center of discussion, while very few studies highlighted the impact of health crises on the wider school populations such as teachers (*n* = 5 studies), school administration (*n* = 1 study), and parents (*n* = 1 study). Although not many studies have been conducted to measure teacher and parent well-being during health crises, a prior study has shown that the implementation of stay-at-home restrictive measures during the pandemic could lead these groups to face serious problems such as pandemic-related stress, depression, anxiety, domestic violence, divorce, and pregnancy [81]. Therefore, further studies and strategic mitigation plans should be carried out to ensure these parties are not left out. With the many changes and negative outcomes surrounding teachers, members of school administration, and parents, their psychological well-being should be considered especially in the uncertainty that comes with school closures and the transition from school closure to school reopening.

This review emphasizes the need to examine strategies that have been used to mitigate the impact of health crises on school populations in future studies, particularly in mental health, teaching and learning, and quality of life. The education system needs to strengthen its mental health and psychosocial support to wider stakeholders beyond students either through in-person or virtual counselling. In terms of teaching and learning, while a few studies highlighted that virtual learning offers flexible, interactive, and effective lesson delivery [82], more studies have revealed that a physical class at school is the most effective delivery modality during the pandemic [83].

The US Center for Disease Control (CDC) has outlined five main mitigation strategies for school reopening, including; (1) consistent and correct use of masks, (2) adherence to social distancing measures, (3) hand hygiene and respiratory etiquette, (4) cleaning and disinfection; and (5) contact tracing in collaboration with the local health department [84]. In order to reduce the capacity of classrooms during school reopening, an earlier study also suggested the multiple cohorts rotation approach, in which in-person and remote classes would be conducted in parallel [85]. Mitigation strategies on the impact of the COVID-19 pandemic on teacher quality of life is also imperative to be addressed. Corroborating the findings of this review, the Organisation for Economic Co-operation and Development (OECD) has confirmed that the teleworking approach which has been commonly utilized during the pandemic was mentally taxing, has increased the risks of burnout and made it difficult for workers to manage their work-life balance [86].

An interesting observation found in this review was that all the studies did not utilize a theoretical framework in investigating factors and the impact of health crises. Theoretical frameworks are useful to understand and explain the relationships between predictors and behavior. Although the reviewed studies identified factors that have significant associations with the impact of health crises, most of the factors were either unable to be manipulated (i.e., gender and income) or were not structured to allow for the effective planning of intervention. Studies utilizing theoretical frameworks may provide a more comprehensive approach to understanding the impacts of health crises.

Several limitations were identified in our review process. First, although this review addressed the impacts and factors surrounding a certain phenomenon, meta-analysis approaches were not used to summarize the empirical evidence. A statistical calculation method could not be conducted to provide a more precise estimate of the impacts of health crises and the associated factors because it was decided to include studies that were carried out qualitatively. Second, this study identified COVID-19 as the only type of health crisis whose impact had been observed among school populations. Therefore, this study suggests that future reviews prolong the timeframe of the study so health crises studied before the year 2000 could also be included and screened. Additionally, as this review only provided information regarding the impact and factors surrounding the health crises, further studies addressing mitigation strategies of the impacts of health crises among school populations are also imperative.

## 5. Conclusions

This study aimed to review the impact of health crises on school populations by examining research conducted between the year 2000 to April 2021. It was found that among the 41 papers examined, the COVID-19 pandemic was the predominant health crisis in studies observing the impacts and its associated factors on school populations. Four notable impacts of COVID-19 were identified: impacts on mental health, teaching and learning, quality of life, and physical health. Among factors associated with the crisis are; demographic factors, followed by concerns about the pandemic, education-related factors, health-related factors, geographic factors, economic concerns, teaching challenges, and parenting in the pandemic. This study is expected to be a reference for future works in formulating crises mitigation strategies to reduce the impact of health crises on schools by exploring the contexts of the crises.

## Figures and Tables

**Figure 1 ijerph-19-04024-f001:**
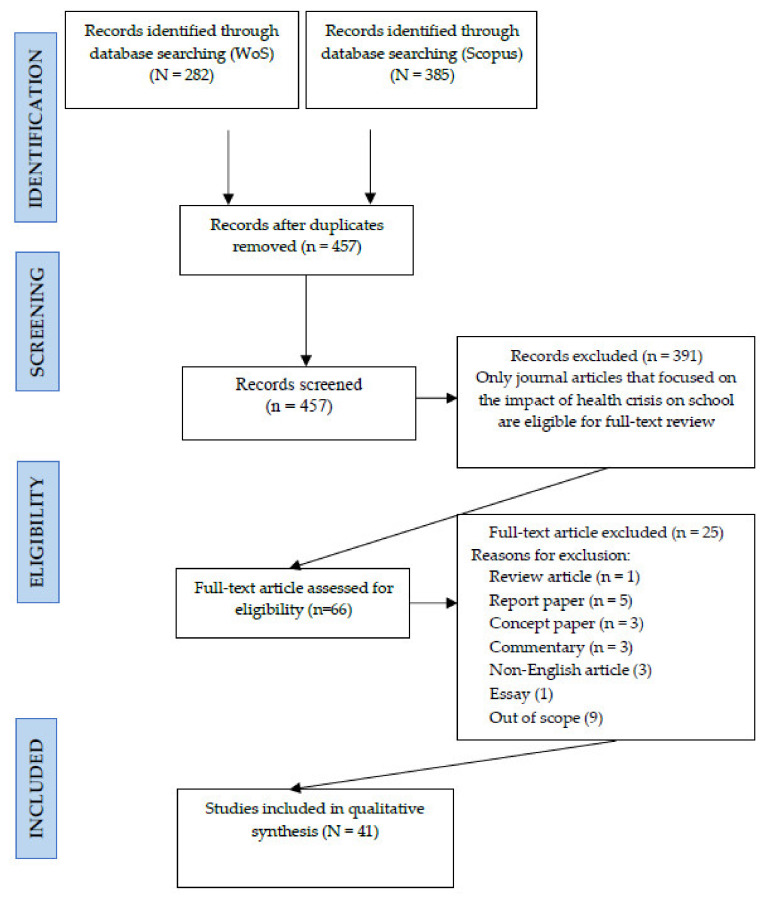
PRISMA flow diagram of the study.

**Table 1 ijerph-19-04024-t001:** Inclusion and exclusion criteria.

Criteria	Eligibility	Exclusion
Document type	Journal article (empirical research)	Review, letter, conference paper, book, book chapter, note, editorial
Source Type	Journal	conference proceeding, book, book series
Language	English	Non-English
Timeline	2000–2021	<2000
Index	SSCI, SCIE, ESCI, A&HCI	Other than specified

**Table 2 ijerph-19-04024-t002:** Keywords search string.

Database	Keywords
Web of Science	TS=((“Health crises” OR “Health Emergenc *” OR “Health disaster”) AND (“Effect*” OR “Impact*” OR “Implication*”) AND (“School*” OR “Teacher*” OR “Student*” OR “Parent*”))
Scopus	TITLE-ABS-KEY((“Health crises” OR “Health Emergenc*” OR “Health disaster”) AND (“Effect*” OR “Impact*” OR “Implication*”) AND (“School*” OR “Teacher*” OR “Student*” OR “Parent*”))

**Table 3 ijerph-19-04024-t003:** Characteristics of the included studies.

No	Author(s)	Year	Country	Population	Method	Impact	Factor
1	Liu et al.	May-20	China	Primary school & college student	Survey	Mental health	Concern about the pandemic
2	Giménez-Dasí et al.	Oct-20	Spain	Primary school student	Survey	Mental health	-
3	Dangal & Bajracharya	Jan-20	Nepal	High school, college, & university student	Survey	Mental health	Demographic, education-related factors, economic concerns
4	Ferraro et al.	Nov-20	Italy	High school student	Survey	Teaching and learning	-
5	Yang et al.	Feb-21	China	High school & University student	Survey	Mental health	Concern about the pandemic
6	Chen et al.	Mar-21	China	High school student	Survey	Mental health	Demographic, education-related factors, health-related factors
7	Schwartz et al.	Mar-21	Canada	High school student	Survey	Mental health, teaching and learning	Demographic, health-related factors
8	Yao Zhang et al.	May-20	China	College student	Survey	Mental health	Concern about the pandemic, health-related factors
9	Yan Zhang et al.	Jul-20	China	College Student	Survey	Mental health	Geographic factors, concern about the pandemic
10	Tasso et al.	Oct-20	US	College student	Survey	Mental health	Concern about the pandemic, education-related factors
11	George & Thomas	Nov-20	India	College student	In-depth interview	Mental health	Concern about the pandemic
12	Mok et al.	Nov-20	China & Hong Kong	College & university student	Survey	Teaching and learning	-
13	Yu et al.	Jan-21	China	College student	Survey	Mental health	Health-related factors, education-related factors, concern about the pandemic
14	Han et al.	Mar-21	China	College student	Survey	Mental health	Demographic, concern about the pandemic, education-related factors, personality traits
15	Marelli et al.	Jul-20	Italy	University student & Administration staff	Survey	Mental health	Demographic
16	de Maio Nascimento	Jul-20	Brazil	University student	In-depth interview	Mental health, physical health	Concern about the pandemic, economic concern
17	Park et al.	Aug-20	South Korea	University student	Survey	Teaching and learning	-
18	Sáiz-Manzanares et al.	Aug-20	Spain	University student	Survey	Teaching and learning	-
19	Hung et al.	Sep-20	US	University student	Survey	Teaching and Learning	-
20	Rana et al.	Sep-20	US	University student	Survey	Mental health, teaching and learning	-
21	Drissi et al.	Oct-20	UAE	University student	Survey	Mental health	Demographic
22	Beery	Oct-20	Sweden	University student	Action research	Teaching and learning	-
23	Song et al.	Oct-20	China	University student	Survey	Mental health	Geographic factors, health-related factors, economic concerns, education-related factors
24	Tejedor et al.	Oct-20	Italy, Spain & Ecuador	University student & lecturer	Survey	Teaching and learning	Education-related factors
25	Saddik et al.	Nov-20	UAE	University student	Survey	Mental health	Demographic, concern about the pandemic, education-related factors
26	Kamil et al.	Nov-20	Sudan	University student	Survey	Mental health	Education-related factors
27	Tasnim et al.	Nov-20	Bangladesh	University student	Survey	Mental health	Demographic, health-related factors, education-related factors
28	Izagirre-Olaizola & Morandeira-Arca	Dec-20	Spain	University student	Survey	Teaching and learning	-
29	Karasmanaki & Tsantopoulos	Dec-20	Greece	University student	Survey	Mental health	-
30	Bourion-Bédès et al.	Jan-21	France	University student	Survey	Mental health	Demographic, concern about the pandemic
31	García-Alberti et al.	Feb-21	Spain & Peru	University student	Survey	Teaching and learning	Education-related factors
32	Conrad et al.	Feb-21	US	University student	Survey	Mental health	Geographic factors
33	Cam et al.	Mar-21	Turkey	University student	Survey	Mental health	Demographic
34	Alghamdi	Mar-21	Saudi Arabia	University student	Exploratory qualitative	Teaching and learning	
35	Podlesek & Kavcic	April-21	Slovenia	University Student	Survey	Mental health	Concern about the pandemic, health-related factors, personality traits
36	Race	Jul-20	Philippines	Schoolteacher	Survey	Teaching and learning	-
37	Prado-Gascó et al.	Sep-20	Spain & Mexico	Schoolteacher	Survey	Mental health	Teaching challenges
38	Lizana et al.	Apr-21	Chile	Schoolteacher	Survey	QoL	Demographic
39	Rabacal et al.	Oct-20	Philippines	College lecturer	Survey	QoL	Demographic
40	Fontanesi et al.	Jun-20	Italy	Parents	Survey	Mental health	Parenting in the pandemic
41	Zaccoletti et al.	Dec-20	Italy & Portugal	Parents	Survey	Teaching and learning	Demographic, education-related factors

## Data Availability

Not applicable.

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
