# Peer review of "Impact of COVID-19 on School Populations and Associated Factors: A Systematic Review"

_ijerph, 2022, doi:10.3390/ijerph19074024_

Round 1
Reviewer 1 Report
The peer-reviewed study addresses a very important issue regarding the impact of the Covid-19 pandemic and preventive measures taken by public authorities, mainly on the well-being of students and other school stakeholders in many countries. The results of the study were derived from the content of 41 peer-reviewed scientific articles published in English. The authors of the study refer to the population around the world.
In relation to the presented description of the study, the following observations arise:
1) the subject of the study does not reflect the content of the study; affects many types of health crises;
2) the study has been excessively extended with tables and figures, the content of which is well described in the text;
3) different terms were used in relation to the research subject: school and related population, ecosystem, stakeholders of the education system;
4) the presented results are dominated by the description of the situation of the academic community;
5) the contents of lines 212 to 225 may be presented in the form of a table;
6) the analysis of the research results is not structured in relation to the levels of education.
The above-mentioned weaknesses of the study make the authors reconsider such issues as: the subject of the study, its scope and content, and the method of analyzing the results.
Reviewer 2 Report
This is a fairly polished article with good presentation, and I liked the thorough and systematic approach to the review employed by the authors.
I thought that the reduction from 391 to 66 papers was dealt with a little briefly and risked being rather subjective. I appreciate that there needs to be a manageable number of items in the review but I think there needs to be more detail on this. Were the papers accepted for review clustered around some of the search terms, for instance, or did they come primarily from one search engine?
It also wasn't that clear to me why certain reports or commentaries were excluded.
Seeing as only Covid-19 related papers were selected I would consider changing the title of the paper to reflect this.
The summary on pages 8-12 of the manuscript is clear but I would not describe it as particularly synthetic! In fact the synthesis is in the discussion section. It is quite clear although perhaps would benefit from being a bit longer/more detailed. I think it would also be advisible to make the connections with the 41 papers clearer by citing them more frequently in this section.
The conclusion is quite brief. It is claimed that "This study revealed that the COVID-19 pandemic was the only health crisis examined in studies observing the impact of health crises on school populations" but this is only true of the 41 papers deemed to have been relevant. Are there really no papers about other forms of health crisis? I would look again at the way this has been phrased. I would also encourage the authors to do a little more to develop their presentation of the overall trends. Essentially there is a list of factors at present but some indication of their relative importance in different contexts or changes over time, for instance, would be very welcome.
